# Differential Effects of Fundamental and Longitudinal Life History Trade-Offs on Delay Discounting: An Evolutionary Framework

**DOI:** 10.3390/bs12030063

**Published:** 2022-03-01

**Authors:** Junsong Lu, Qi’an Lu, Lin Lu

**Affiliations:** School of Economics and Management, Guangxi Normal University, Guilin 541004, China; lulin355@163.com

**Keywords:** delay discounting, intertemporal choice, age differences, life history theory, time perception

## Abstract

We synthesized life history theory and the antagonistic pleiotropy hypothesis to form an integrative framework for understanding delay discounting (DD). We distinguished between fundamental and longitudinal life history trade-offs to explain individual and age differences of DD. Fundamental life history trade-offs are characterized by life history strategies (LHS), describing how individuals adjust reproductive timing according to childhood environments, while longitudinal life history trade-offs characterize how individuals make trade-offs between early- vs. late-life reproduction as a function of age. Results of a life-span sample (242 Chinese participants) supported several theoretical predictions: (a) slower LHS predicted lower DD; (b) the relationship between chronological age and DD was U-shaped; (c) the effects of age and LHS were differential. Mechanisms underlying fundamental and longitudinal trade-offs were explored. Regarding fundamental trade-offs, LHS mediated the effects of childhood environment on DD. Regarding longitudinal trade-offs, the U-shaped relationship was more evident between physical age and DD: older adults who were in poorer physical health felt older and exhibited a higher DD. Neither the time perspective nor anticipatory time perception mediated the effect of life history trade-offs. We concluded that DD was a product of two distinct life history trade-offs, reflecting both the trait-like quality and age-related development.

## 1. Introduction

Smaller but sooner rewards are normally preferred over larger but later rewards in intertemporal decisions requiring a trade-off between securing resources today and preparing for tomorrow. Preference in intertemporal decision-making is described using delay discounting, indicating to what extent delayed rewards are devalued [1]. As early as two millennia ago, a fable in Chuang Tzu documented this phenomenon [2]: A keeper of monkeys once ordered concerning the monkey’s rations of acorns that each monkey was to have three in the morning and four at night. But at this the monkeys were very angry. So the keeper said that they might have four in the morning, but three at night. With this arrangement, all monkeys were well pleased. Studies in behavioral economics and psychology today authenticated the phenomenon and offered substantial empirical evidence that delayed rewards are often underestimated [3,4,5].

Determinants of delay discounting are still under debate; relevant studies exploring psychological underpinnings of delay discounting can be divided into two broad classes: one stream emphasizes that temporal discounting is inherited and reflects individual tastes [6,7], while the other suggests that since the value of a reward is based on when (e.g., at what age) it is received, delay discounting is determined by maximizing utility of temporal outcomes [8,9,10]. Studies on individual differences demonstrated that delay discounting was largely accounted for by genotype (e.g., heritable) [11,12] and associated with various traits such as intelligence, personality, self-control, and time perspective [13,14,15,16], while the second class is represented by research on the age effect of delay discounting and has been paying much attention in the past three decades [17,18,19,20]. Although both individual and age differences in delay discounting have been extensively studied, current works are distributed across disciplines and were not integrated. Moreover, several fields, especially studies on age differences, demonstrated equivocal results, making few firm conclusions warranted. 

In order to build bridges across disjoint fields and provide explanations for heterogeneous findings, the present study proposed an integrative framework for understanding delay discounting. Building on previous efforts to explain delay discounting from an evolutionary perspective [21,22,23], we synthesized life history theory with the antagonistic pleiotropy hypothesis, proposing that delay discounting was affected by two different life history trade-offs. Specially, we distinguished longitudinal life history trade-offs from fundamental ones. Fundamental life history trade-offs describe how individuals shift reproductive timing as a response to childhood ecological cues, while longitudinal life history trade-offs characterize how individuals make trade-offs between early- vs. late-life reproduction. In the remainder of this introduction, we reviewed life history theory and the antagonistic pleiotropy hypothesis. We then proposed our theoretical framework as well as corresponding predictions, which were then tested in a life-span sample.

### 1.1. Life History Theory and Individual Differences in Delay Discounting

All organisms meet the fundamental challenge to efficiently allocate limited energy and resources across different activities. Life history theory, a middle-level theory of evolutionary biology, distinguishes somatic effort that invests in survival-enhancing activities (e.g., maintenance and growth) from the reproductive effort that invests in reproduction and mating [24,25,26]. The trade-off between the two efforts results in an organism’s life history strategy (LHS), with fast life history strategists allocating more resources to reproductive efforts, preferring earlier timing of puberty and reproduction, while slow life history strategists mature at a slower rate, preferring late onset of reproduction. This fundamental life history trade-off mirrors intertemporal decision-making [23,27]: somatic effort is analogous to saving behavior and thus improves organisms’ reproductive success in the future, while reproductive effort is analogous to spending and thus aims to replicate one’s gene as soon as possible. 

Additionally, life history theory predicts that the major evolutionary impetus of LHS is an organism’s childhood environment [21,22]. Specifically, LHS is contingent on harshness (e.g., mortality rate), unpredictability, and resource scarcity of the childhood environment. In an environment of high levels of harshness and unpredictability, organisms benefit less from somatic effort; only maturing and reproducing at a relatively young age would they seize the opportunity to propagate their genes. Conversely, when the environment is stable and less harsh, natural selection favors a slower strategy because investing more in somatic effort under such conditions increases life expectancy and could fulfill their full reproductive potential. 

This perspective indicates that psychological processes are adapted to ecological cues encountered in early development and that LHS underpins the calibrating mechanism [28,29]. Given the environmental contingency of LHS, it is now becoming evident of the relationship between LHS and delay discounting. Individuals exposed to mortality cues in childhood implicitly infer that environmental harshness and unpredictability are high. They are therefore aware that they are unlikely to benefit from waiting for a larger but delayed reward. By contrast, if individuals grow up in stable environments and expect themselves to live longer, they become more future-oriented and more likely to choose a delayed reward. This argument has received support from experimental research [23,30], suggesting that childhood SES serves as an important predictor of “the personal taste” of time preference.

### 1.2. The Antagonistic Pleiotropy and Developmental Trajectory of Delay Discounting

Although life history theory describes the development of LHS, it mainly focuses on early childhood and is used to explain individual differences. It helps predict and explain between-subjects variance in delay discounting but is not explicit about its lifelong development. 

Taking a similar life history trade-off perspective, the antagonistic pleiotropy hypothesis of senescence [31] provides a theoretical basis for studying the age effect on delay discounting. It assumes that antagonistic genes exist in nature, and an individual’s chances of reproducing decrease with age. Pleiotropic genes are genes that have opposing effects on fitness (e.g., reproductive success) at different ages. The antagonistic pleiotropy hypothesis predicts that natural selection would favor pleiotropic genes that are beneficial in early life but deleterious later in life. Once these deleterious effects begin to accumulate, senescence takes root. Thus, senescence could be viewed as a trade-off between early and later life reproductive success. Regardless of mortality, the probability of surviving to age T + 1 is always lower than that of age T, so it is always optimal to sacrifice reproductive opportunities in later life for opportunities for early reproduction after reaching sexual maturity. In a realistic setting, the hypothesis implies accelerated senescence. The later a deleterious effect occurs, the less it affects overall reproductive success. Therefore, a pleiotropic gene with larger but later deleterious effects could be selected. Although no genes of the sort hypothesized by the theory were found in the 1960s, genetic evidence supporting antagonistic pleiotropy has been accumulating in recent years (for a review, see [32]). 

From this perspective, when examining age differences in delay discounting, one should explore how a fitness-relevant reward is discounted in the context of accelerated senescence. Organisms that have experienced a significant decline in physiological function are unlikely to be benefited by waiting for a delayed reward. This predicted an increase in delay discounting at old ages. In contrast, in early life, before senescence begins, the benefits of pleiotropic genes gradually manifest with sexual maturation. If the organism survives as time goes by, a lower external mortality rate is more likely and suggests a decline in survival rate at a decreasing rate [9]. Because the reproductive probability is a function of fertility and survival probability, this means that more reproductive opportunities will be available in the future and that waiting for delayed rewards can maximize fitness.

Where there is a sex difference, the antagonistic pleiotropy hypothesis predicts that selection operates weakly on females since their reproduction mostly occurs before menopause [33,34]. However, this does not imply that women will not experience an increase in delay discounting later in life. First, senescence continuously increases intrinsic mortality rates and makes delayed rewards less likely to be received. Second, with a relatively long post-menopausal life-span, women can invest in grandchildren to increase inclusive fitness [35]. This ability to invest must gradually decline with aging, and it is better to obtain resources earlier.

### 1.3. Delay Discounting as the Product of Fundamental and Longitudinal Life History Trade-Offs

Based on our discussion on life history theory and the antagonistic pleiotropy hypothesis, we proposed a distinction between fundamental and longitudinal life history trade-offs and argued that delay discounting is the product of these life history trade-offs. Fundamental life history trade-offs, or life history strategies, predict how individuals respond to ecological cues (e.g., mortality rate) and thus explain individual differences in delay discounting. Individuals growing up in higher socioeconomic status (SES) environments generally have lower delay discounting and vice versa. Based on the antagonistic pleiotropy hypothesis, longitudinal life history trade-offs predict developmental trajectories of delay discounting. Individuals would show an increase in delay discounting as senescence progresses, regardless of their life history strategies. Early life, in contrast, should witness a decrease in delay discounting as sexual maturity approaches, as higher reproductive probability will be achieved in the future.

Fundamental and longitudinal life history trade-offs are differential underpinnings of delay discounting for several aspects. First, fundamental trade-offs explain the between-subjects variance of delay discounting, while longitudinal trade-offs account for the within-subjects variance. Both mechanisms should be taken into consideration when comparing delay discounting across individuals and groups, as one of them could blur the effect of the other. Previously inconsistent results on the age effect of delay discounting [36,37,38] may therefore be due to the failure to control for LHS or childhood SES. Second, fundamental and longitudinal trade-offs differ in their evolutionary impetus. Fundamental trade-offs are contingent on childhood SES [21], while longitudinal trade-offs are driven by adult mortality rates [31]. Therefore, even if both adopt a trade-off perspective, their influences are independent.

One might question the legitimacy of applying evolutionary theory in delay discounting research because inferences about delay discounting are all based on fitness-relevant rewards, and most studies use monetary rewards which did not exist during evolutionary history. Further, the extent to which the inferences of evolutionary theory are universal is still in question. We argued that the predictions above could be applied to delay discounting of monetary reward. People’s desire for money is a modern derivative of the desire for bioenergetic resources (e.g., food) because the need for fitness-relevant resources is found to be consistent with the thirst for money [39,40]. We also argued that these predictions could be extended to different types of rewards. Overwhelming evidence indicated that cross-domain (e.g., food, money, health, and alcohol) discounting rates were strongly correlated [6,41,42,43]. If food and money are important for survival and mating, then related resources or resources with similar discounting rates would also be related to fitness.

### 1.4. The Current Study

Previous studies have examined the effect of childhood SES on delay discounting (e.g., [23]) as well as age differences (for a review, see [44]). The current study integrated these two traditions in an evolutionary perspective to facilitate the understanding of delay discounting. Based on the framework, several hypotheses were tested. 

The first hypothesis is that LHS should systematically influence delay discounting at all ages. A faster life history strategist would prefer smaller but sooner rewards more than a slower strategist. This fundamental life history trade-off follows the pathway from childhood SES to LHS and then to delay discounting. Second, longitudinal life history trade-offs predict a U-shaped relationship between age and delay discounting. These two trade-offs serve as different distal factors and should independently influence delay discounting. We also expected the sex with higher mortality rates and underwent more rapid senescence (males) to show higher delay discounting. The aforementioned variables were measured. Considering that people age at different rates, we also measured subjective ages and their related subjective health conditions, both physical and psychological.

One aspect was not explicit in the framework: what are the proximal mediators of life history trade-offs. Based on the literature, the experience of time may underlie age and individual differences in delay discounting. For example, future time perspective was associated with delay discounting [13]. However, it is unclear whether time perspective serves as a mediator of fundamental life history trade-offs or is merely a life history indicator. Regarding age differences, although future time perspective has been reported as a mediator of age among adolescents and young adults [45], the result was not replicated subsequently [19] and requires further exploration. To provide an initial insight into potential mediation effects of time, we measured future time perspectives. Considering the lack of evidence for the role of time perception in delay discounting and how it connects with life history trade-offs, we also measured anticipatory time perception, which is a more relevant measure of time perception in the delay discounting tasks [46].

## 2. Materials and Methods

### 2.1. Participants

Sample sizes were determined to be comparable or greater than previous studies that detected an age difference in delay discounting [19,47]. Two hundred and forty-two Chinese participants (age: M = 42.44, SD = 15.75, Range = 18 to 69) were recruited from two online platforms, Credamo and Weidiaocha. The samples of both platforms are universal, covering the populations of all provinces and socioeconomic status in China. Based on the distribution, participants were classified as young adults (age range 18–33), middle-aged people (36–46), or older adults (52–69). All participants had completed the tasks while passing the attention check. Following completion of all tasks, they were compensated 4 to 6 RMB, depending on the platform they registered. Although the incentive is a fixed amount, convergent evidence indicates that experiments without performance-based incentives (e.g., rewards based on choices in delay discounting tasks) are still justified [48]. The ethical standards and procedures for research with human beings were met and approved by Guangxi Normal University (approval number: LQ001). Demographics for each group were summarized in Table 1.

### 2.2. Measurements

#### 2.2.1. Age

In addition to chronological age, participants reported two additional subjective ages: subjective physical and psychological age. Specifically, they responded to “how old do you feel physically/psychologically” in an open-ended question, which followed previous studies [49,50]. 

#### 2.2.2. Subjective Health

Corresponding to age, participants reported their physical and psychological health on two 5-point items: “How would you describe your current physical/psychological health?”. A higher score indicated better subjective health. These items have been used in previous studies and showed good validity [49]. 

#### 2.2.3. Delay Discounting 

Delay discounting was assessed using 7 intertemporal choices adapted from previous studies [51,52]. In each choice, participants make decision between an immediate reward (range from 840 RMB to 3990 RMB) and a larger reward (range of 2311 RMB to 8190 RMB) after a delay (range from 4 to 939 days). Delay discounting rate was indexed by a hyperbolic discount parameter *k* that best summarized the decisions of a participant. A higher *k* indicates a stronger preference for smaller but sooner rewards. More details about the method for estimating *k* were presented in the methods section.

#### 2.2.4. Future Time Perspective

Individual differences in time perspective were measured by the future time perspective (FTP) scale. Participants rated 10 items about to what extent they focus on opportunities and limitations in the future from 1 (very untrue) to 7 (very true), with higher scores indicating more future-oriented and expansive time perspective. Internal consistency was examined using coefficient omega, which is a more appropriate reliability estimate than Cronbach’s alpha [53]. The internal consistency was good, coefficient omega = 0.88, 95% CI [0.85, 0.90].

#### 2.2.5. Anticipatory Time Perception

In ten slider questions, participants were asked to mark how long they feel from today to a certain point in the future by dragging the bar in a 150-mm line. The left and right ends of the line were labeled “very short” and “very long”, respectively. The time horizons ranged from 3 to 60 months and were randomly presented. 

#### 2.2.6. Life History Strategy

The Mini-K was used to determine life history strategies of participants. The Mini-K is a 20-item short-scale derived from the Arizona Life History Battery [22,54]. Sample items include “while growing up, I had a close and warm relationship with my biological mother”, and “I have to be closely attached to someone before I am comfortable having sex with them”. A higher total score suggested a slower life history strategy. The reliability of the Chinese version of the Mini-K was good, omega = 0.88, 95% CI [0.86, 0.91]

#### 2.2.7. Childhood Socioeconomic Status

Three childhood SES indicators [23] that evaluate family economics were measured. Participants rated items on a 7-point scale from 1 (strongly disagree) to 7 (strongly agree). A sample item is “My family usually had enough money for things when I was growing up “. The internal consistency is high, omega = 0.91, 95% CI [0.89, 0.93].

### 2.3. Analytical Strategy

Before data analysis, the process under which our theoretical framework was constructed and hypotheses were formulated required further clarification. The current study adopted a hypothetico-deductive model that follows deductive reasoning [55]. Scientific inquiry begins with formulating hypotheses based on existing theories. These hypotheses are then tested with observable data when the results are not yet known. In evolutionary psychology, the construction and testing of theories are subjected to such a process: while following the theory of evolution, middle-level theories (e.g., life history theory) generate falsifiable hypotheses and predictions that are directly tested by empirical data [56]. Therefore, our qualitative review is constructed in theories and our quantitative analyses provide evidence for hypotheses derived from qualitative theories. 

Two aspects warrant attention in data analysis. First, multiple outliers could be identified while addressing behavioral measures, particularly delay discounting. Classical statistics are not robust to outliers and could result in extremely misleading outcomes (e.g., Pearson’s product moment correlation). Second, nonnormality, typically exhibited in Likert scales and behavioral measures, poses a serious problem for classical statistics [57,58]. A slight departure from normality drastically reduces power and biases results of classical statistical methods [58,59]. Normality of our measures was tested using Shapiro-Wilk’s test. All *p* values were smaller than 0.011, indicating a severe departure from the normal distribution and validated the use of robust statistical methods.

To address these issues, most statistical inferences in this study used robust counterparts of classical statistics. To identify associations, Kendall’s tau, a rank-based correlation, was used. To more accurately measure locations, an M-estimate of location [60] rather than a mean was used. For fitting linear regression, the MM-estimator [61], rather than the ordinary least squares estimator, was used. For detecting mediation effects, a robust mediator model proposed by Zu and Yuan [62] was used. The implementation of the aforementioned techniques followed the instructions of Mair and Wilcox [63] using the R package “WRS2” and “robustbase” [64].

## 3. Results

### 3.1. Preliminary Analysis

#### 3.1.1. Delay Discounting

Following the well-established tradition, the delay discounting rate was represented by *k* in a hyperbolic discounting function [65]:SV = A/(1 + *k*D),(1)
where A is the delayed reward, D is the delay in days, and *SV* refers to the participant’s subjective value of the delayed reward after it is discounted to the present day. A higher *k* represents a higher delay discounting rate.

In order to estimate *k* for each participant, choice data (choices of the seven intertemporal decisions) for each individual was fitted using maximum likelihood estimation. Under the Luce choice rule [66], we defined the likelihood function of a single choice as:(2)$$P(k,\;\sigma )=\{\begin{array}{c} \frac{1}{1+e^{-\sigma \left(SS-SV\right)}},\;if\;SS\;is\;selected\\ 1-\frac{1}{1+e^{-\sigma \left(SS-SV\right)}},\;if\;LL\;is\;selected \end{array}$$
where *P* is the probability of choosing a particular option, *SS* is the magnitude of the immediate reward, σ is the parameter that measures to what extent choice is guided by the difference between the values of the two options. Data for seven participants failed to converge and were excluded for analyses, including delay discounting. All *k* values were then log-transformed.

#### 3.1.2. Anticipatory Time Perception

Kim and Zauberman [46] demonstrated that people exhibit diminishing sensitivity to the perception of longer time horizons. We fitted time estimate data of each participant with a nonlinear function, using the robust least squares estimator:(3)$$T=\alpha t^{\beta },$$
where T is the subjective time perception in months, *t* refers to the calendar time in months. The two parameters, α and β, capture the overall level of time contraction and diminishing sensitivity to calendar time, respectively. A higher α indicates a high level of contraction, and a higher β indicates a higher insensitivity to time. The lower and upper bounds of the parameters were set to zero and five, respectively. Observations where parameters reached bounds were treated as outliers; outliers were further detected by boxplots. A total of 47 participants were excluded from the analysis of time perception.

### 3.2. Delay Discounting as Fundamental and Longitudinal Life History Trade-Offs

Participants who grew up in higher SES environments generally took slower life history strategies, with Kendall’s τ = 0.22, *p* < 0.001. At all ages, life history strategy was negatively associated with delay discounting, with Kendall’s τ = −0.15, *p* = 0.002. 

Then, we calculated Kendall’s tau between chronological age and delay discounting for each group. A graphic depiction of these associations is shown in Figure 1. For young adults, delay discounting decreased with age, τ = −0.23, *p* = 0.005. This association was reversed for older adults, τ = 0.15, *p* = 0.051. Interestingly, physical age resulted in even stronger associations with delay discounting for both young adults, τ = −0.27, *p* = 0.001, and older adults, τ = 0.17, *p* = 0.028. This reversal was not held for psychological age (young adults, τ = −0.18, *p* = 0.029; older adults, τ = 0.09, *p* = 0.34). There was no significant association between chronological age and delay discounting in the middle-aged group, τ = −0.03, *p* = 0.78.

Sex differences were examined. Across age groups, only young males exhibited higher discounting rates than young females after controlling for life history strategy, *F*(1, 80) = 4.15, *p* = 0.045. As predicted, the influence of senescence is stronger among old males (τ = 0.20, *p* = 0.043) than females (τ = 0.10, *p* = 0.44). However, the interaction between sex and age failed to reach significance, *b* = 0.002, *t* = 0.25, *p* = 0.80.

### 3.3. Differential Effects of Fundamental and Longitudinal Life History Trade-Offs on Delay Discounting

Differential effects of fundamental and longitudinal life history trade-offs have already been implied in their associations with delay discounting: life history strategy exerting a linear influence for all participants, whereas the age effect followed a U-shaped pattern.

To provide further evidence for the divergent effects of age and life history strategy on delay discounting, we regressed delay discounting on life history strategy, chronological age, age group (dummy coded, zero for young adults and one for older adults), and the interaction between age and age group in a robust regression (Model 1). Note that only young and older adults were included. According to our theoretical framework, the change in fertility is more pronounced around sexual maturity and at older ages. Thus, the contrast between these two periods (young and older adults) deserves more attention. The results are shown in Table 2. Life history strategy significantly predicted delay discounting after controlling for the age effect. The interaction term was also significant; chronological age was positively associated with delay discounting only for older adults.

### 3.4. Mechanisms Underlying Fundamental Life History Trade-Offs

Mediation effects were tested by estimating robust mediator models [62]. We defined a mediation effect as the product of the regression coefficient of the X (the predictor) to M (the mediator) path and the coefficient of the M to Y (the outcome) path, following the definition of Preacher and Hayes [67]. Under this definition and procedure, the mediation effect was tested directly.

The robust mediator model showed a significant indirect effect of childhood SES on delay discounting via life history strategy, a × b = −0.09, bootstrap 95% CI [−0.16, −0.03]. Neither future time perspective nor anticipatory time perception served as significant mediators (all *p*s > 0.09). Subsequent analyses found that future time perspective was significantly associated with life history strategy, τ = 0.35, *p* < 0.001, but not delay discounting, τ = −0.07, *p* = 0.14. This suggested that future time perspective was more likely to be a life history indicator. Moreover, there was no robust association between life history strategy and time perception parameters (all *p*s > 0.14). 

### 3.5. Mechanisms Underlying Longitudinal Life History Trade-Offs

In all three groups, no significant associations between chronological age and future time perspective were found (see Table 1). Combining the result that future time perspective was not associated with delay discounting, its potential mediation effect was not examined. 

Next, we investigated whether age systematically affected anticipatory time perception. As a graphical depiction, subjective time was plotted against calendar time for all three groups in Figure 2. The figure indicated minor age differences, even when the delay was large. 

Associations between chronological age and time perception parameters for each group were shown in Table 1. Previous studies [46,68] suggested that an increase in alpha and a decrease in beta were associated with higher discounting rates. If so, developmental trajectories of time perception parameters and delay discounting were consistent among young adults. Thus, decreasing alpha and increasing beta with age may explain the negative age effect on delay discounting among young adults. However, two additional findings indicated that this argument needed to be taken with a grain of salt. First, though the trends in the associations between delay discounting and time parameters were consistent with those of Kim and Zauberman’s study [46], these two associations failed to reach statistical significance in the total sample (alpha, τ = 0.07, *p* = 0.20; beta, τ = −0.06, *p* = 0.27). Second, age differences in alpha and beta did not predict developmental trajectories of delay discounting among middle-aged and older adults, which indicated that anticipatory time perception was not an important factor for determining delay discounting over life-span.

So far, the underlying mechanism under which chronological age affected delay discounting remained unclear. Given the association between physical age and delay discounting, we hypothesized an incremental effect of age bias (discrepancy between subjective age and chronological age) on delay discounting, which may further indicate a potential association between health conditions and delay discounting. Therefore, we calculated age bias scores by subtracting chronological age from physical age. A positive score indicated that the participant believed him or that she was older than he or she actually was. In Model 2 (see Table 2), there was a significant interaction between age bias scores and age group after controlling the age effect. Among older adults, participants who subjectively felt they were older physically exhibited higher discounting rates. This interaction was insignificant for age bias score based on psychological age (*b* = 0.06, *t* = 1.48, *p* = 0.14). The results unraveled the possibility that physical health conditions might underpin the possible regulatory mechanism behind the age-delay discounting association, as people might feel they are older because of their poor health conditions. Indeed, subjective physical health negatively correlated with age bias scores at all ages, τ = −0.21, *p* < 0.001. Thus, we hypothesized that older adults who were in poorer physiological conditions felt that they were older and thus were more preferred smaller but sooner rewards. As predicted, the indirect effect of physical health to delay discounting via physical age was significant (Figure 3), a × b = −0.13, bootstrap 95% CI [−0.32, −0.001]. In contrast, delay discounting was not driven by physical health among young adults, a × b = −0.14, bootstrap 95% CI [−0.37, 0.03], nor among middle-aged, a × b = 0.059, bootstrap 95% CI [−0.14, 0.26]. 

## 4. Discussion

From an evolutionary perspective, we synthesize life history theory and the antagonistic pleiotropy hypothesis of senescence, providing an integrative theoretical framework for understanding delay discounting. The framework helps integrate existing disjoint research and makes clear predictions about individual, age, and sex differences. We then provide empirical evidence for differential effects of fundamental and longitudinal life history trade-offs on delay discounting, which are the two distal causes, and explain the trait-like quality and age-related development of delay discounting. Their proximal mechanisms were then explored, especially the role of future time perspective and anticipatory time perception.

### 4.1. Fundamental and Longitudinal Life History Trade-Offs

As predicted, LHS is contingent on childhood SES, and slower LHS is associated with lower delay discounting in the total sample. This squares with previous experimental works showing that mortality cues influenced time preference as a function of childhood SES [23,30]. Our analyses push the literature further by demonstrating the mediation effect of LHS in a robust mediator model and providing support for fundamental life history trade-offs. At the outset of this paper, we presented the fable in *On the Equality of Things* by Chuang Tzu [2]. Chuang Tzu used this story to tell that regardless of having three in the morning and four at night or having four in the morning and three at night, they are “equal” and are two sides of the same coin. In line with life history theory, whether taking a fast or a slow life history strategy, it is for better survival and reproduction in a certain environment. Slow strategists who grew up in environments with low levels of harshness and expected to live longer are more likely to wait for larger but later rewards. In contrast, faster strategists are not able to wait for large payoffs in the future because they may not live to see tomorrow due to a harsh environment [9,23]. This implies that humans have no intrinsic preference for smaller but sooner rewards. 

Consistent with previous studies [20,69,70,71], we found a U-shaped relationship between chronological age and delay discounting predicted by longitudinal life history trade-offs. Different from previous studies, we made a theoretical explanation for the observed age difference and linked it with other studies in an integrative framework: The U-shaped relationship reflects how individuals make trade-offs between early- vs. late-life reproduction as a function of chronological age. However, we do not argue that a U-shaped relationship should be detected for any population or for any environment. A decrease in delay discounting is to be found in early life (not yet sexually mature or just reaching sexual maturity) only when there are more reproductive opportunities in the future than in the present. If an increase in delay discounting is to be found in older ages, advancing age should result in accelerated senescence. Consider a special case in which survival rates drop sharply in young individuals. Even if fertility increases with age, this increase does not compensate for the accelerated decline in survival rates and thus would not result in a decline in delay discounting. However, a more common scenario should be a declining survival rate at a decreasing rate because if the environment is uncertain, the longer one survives, the less likely there will be a greater external hazard in the future [9] and thus indicates greater reproductive success in the future. Combined with accelerated senescence, we suggest that the U-shaped relationship is prevalent.

Sex differences also extract insights into the psychology of delay discounting. Across sexes, the mortality gap reaches its maximum around the age of 20–30 [72] and then gradually decreases. This may explain why we only observed significant sex differences in the younger group. A higher mortality rate prompts young males to seize resources rapidly. Among old adults, males exhibited a greater age-related increase in delay discounting, as indicated by a more positive Kendall’s tau. This discrepancy is not strong enough to achieve a significant interaction. We noticed that old adults showed a significantly lower physical and psychological age than their chronological age (all Cohen’s *d* > 0.72). Old adults who are able to fill out online questionnaires are more likely to come from wealthier families and experience lower rates of senescence, which can weaken the sex difference. 

Another important finding is that the effects of fundamental and longitudinal life history trade-offs are independent. This is attributed to the fact that they are driven by childhood environment and aging in adulthood [21,31], which are distinctive, and hence their underlying mechanisms need to be investigated separately.

### 4.2. Mechanisms Underlying Fundamental and Longitudinal Life History Trade-Offs

To further explain mechanisms underlying life history trade-offs, potential proximal factors were explored. However, neither future time perspective nor anticipatory time perception serve as significant mediators. 

Future time perspective is more likely to be a life history indicator because of its medium-to-strong association with LHS (expressed with a more common metric of effect size, Pearson’s *r* = 0.45). This makes sense because slow life history strategists are expected to live longer and thus be more future-oriented [22,30]. Future time perspective, while measured in Zimbardo Time perspective inventory [73], was also found to be a valid measure of LHS [74]. However, it has only a weak and insignificant association with delay discounting. Therefore, previous findings of covariance between future time perspective and discounting rates (e.g., [13]) may reflect an association between fundamental life history trade-offs and delay discounting. In contrast to the time perspective, life history strategy does not affect overall time contraction and diminishing sensitivity of time. It is unexpected that individuals who expect to live shorter do not perceive a given delay as longer and is thus harder to wait for. A possible explanation is that LHS may not be reflected in the assessment of objective time that has nothing to do with survival and reproduction; only when the objective delay is perceived as a risk (e.g., not being able to receive a reward after the delay) would time perception be influenced by LHS.

We also explored whether age differences in time perception and perspective contribute to delay discounting. Previous studies regarding age effects on time perception mainly focused on retrospective estimates and at relatively short intervals [75,76,77,78], which are not relevant enough to intertemporal decision-making. We extend the current literature by examining anticipatory time perception at various delays ranging from three months to five years. As a result, no age difference in anticipatory time estimates was found. According to the socioemotional selectivity theory [79,80], older people perceive their future time as more limited than younger adults. Then, a conceivable inference is that older adults would perceive a given time period longer (lower levels of time contraction) and be more sensitive to time (lower levels of diminishing sensitivity) than young adults. It is similar to how ten dollars means nothing for a rich man but a lot for a poor man. Nevertheless, this hypothesis is not supported. In terms of time perspective, in agreement with some studies (e.g., [81]), we find no significant age effect on future time perspective. Although these trends are consistent with previous studies that reported increasing future orientation among young adults [45] but decreasing expansive time perspective among old adults [80], they do not account for age differences in delay discounting. This decoupling of time perspective from delay discount is consistent with a previous study using the same scale [19]. We conclude that the future time perspective serves as a life history indicator that better reflects LHS, which is relatively stable.

So, what are the factors through which chronological age affects delay discounting? Our results suggest that these may be physiological factors. Notably, the U-shaped association between delay discounting and physical age, but not psychological age, is more evident than chronological age. Furthermore, young adults who consider themselves as older than their chronological age exhibit lower discounting rates, whereas older adults exhibit higher discounting rates. Based on the theory, chronological age can be viewed as an observable indicator of two latent variables (survival rate and fertility); and physical age may be a more accurate indicator. Young adults with positive age bias scores could be those who experience a faster increase in reproductive probability, which makes a fitness-relevant reward more valuable in the future. By contrast, older adults who perceive themselves as older could be those aging at a faster rate and thus have worse physical health. They, therefore, know they have no future and are more likely to have high delay discounting rates.

### 4.3. A Reinterpretation of Relevant Research

The framework provides insights on explaining current findings of delay discounting and what are potential confounding variables in delay discounting research.

First, we argue that some associations between delay discounting and individual characteristics are confounded by LHS. For example, delay discounting has long been regarded as a measure of self-control [3,82,83]. However, these associations are weak [14,16]. Based on life history theory, both impulsivity and delay discounting are indicators, or consequences, of LHS [22,23]. Therefore, LHS may confound the relationship between self-control measures (e.g., executive functions and impulsivity) and delay discounting.

Our theoretical framework also has implications for age studies. Again, consider the relationship between self-control and delay discounting. Robust evidence supported an increase in self-control [84,85] and a decrease in delay discounting [86] among adolescents. However, this might not suggest that delay discounting is an indicator of self-control; rather, the decline of self-control and delay discounting in youth may be two concurrent but different processes. A recent meta-analysis [87] found little evidence for a general inhibition (indicated by various self-control measures) deficit among older adults. Therefore, self-control cannot account for the observed increase in delay discounting among old people, suggesting a dissociation between these two constructs. On the other hand, empirical evidence showed mixed results for the age effect on delay discounting [18,20,38]. A recent meta-analysis indicated that there is no reliable age difference in delay discounting [44]. However, rather than being falsified, our theoretical framework provides several reasons for these inconsistencies. First, meta-analysis is a hierarchical linear model that assumes a linear relationship between variables [88]. In their meta-analysis, the authors [44] did not provide evidence to determine whether this insignificant effect was caused by curvilinear relationships. If the true effect is curvilinear, then it is possible to observe both positive and negative effects based on how age groups are defined, which account for previously equivocal findings. Second, some studies [18,37] examined the age effect by comparing group means of delay discounting. This practice is susceptible to the influence of fundamental life history trade-offs, which are independent of longitudinal trade-offs. A middle-aged group with the average faster strategy could show higher delay discounting than young adults with slower strategies. Third, the starting point for our discussion is to consider rewards as fitness-relevant. If the magnitude of the money in the decision task is small, it may not follow our prediction. An often-overlooked fact is that age is positively related to income, which was reported to be associated with delay discounting [89]. When the monetary rewards are small, then the age-related change in the discounting rate may simply reflect the lower marginal utility of rewards for wealthier people. This hypothesis was not directly tested in the current study due to a lack of adulthood SES measures and relatively large amounts of money being used in delay discounting tasks. However, a more recent study did find that young adults (mean age = 19.42) exhibited higher discounting rates than older adults (mean age = 72.48) for small (less than $100) but not large money [90]. This finding provided support for our hypothesis. If the U-shaped function was only evident for fitness-relevant but not small rewards, then these two age groups could have a comparable discounting rate for large money.

## 5. Limitations and Conclusions

The current study is subject to several limitations. First, as we discussed above, sampling bias may present because we recruited participants from online platforms. China has seen a marked change in the economy over the past several decades, and old adults who are able to use computers are more likely to grow up in high SES environments. Fortunately, this factor is measured and controlled. A second limitation is that although we defined longitudinal life history trade-offs, the current study uses a cross-sectional design. Future studies could test our predictions using a longitudinal design. Third, as mentioned above, delay discounting may also reflect the marginal utility of monetary rewards. In order to rule out this potential confounder, individual differences in wealth need to be controlled. However, as mentioned above, this issue was not serious for medium-to-large money.

In summary, the current study provides a unifying framework for understanding individual and age differences in delay discounting. Most importantly, fundamental and longitudinal life history trade-offs affect delay discounting independently. Furthermore, although delay discounting does appear to have trait-like quality [43], the tendency to discount rewards changes with age and does not reflect a simple development process that is common for personality characteristics [91]. Rather, its developmental trajectory is characterized by longitudinal life history trade-offs that reflect a trade-off between early and late-life reproduction. We emphasize that physical status plays a crucial role in characterizing age-related changes in delay discounting, whereas the effects of cognitive factors are peripheral.

## Figures and Tables

**Figure 1 behavsci-12-00063-f001:**
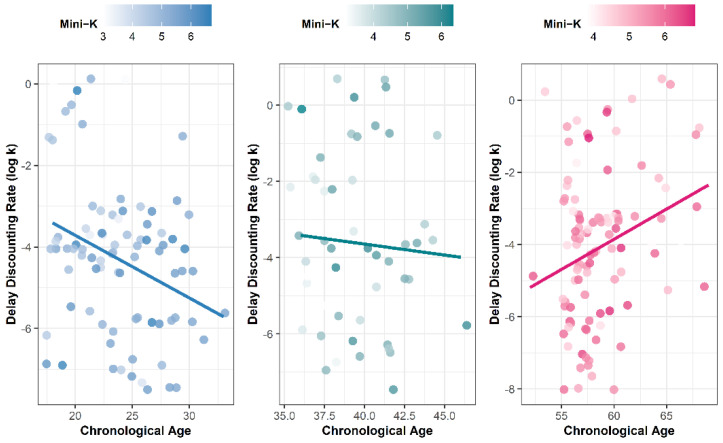
Blue (**left**), green (**middle**) and red (**right**) represent young adults, middle-aged, and older adults, respectively. Each straight line represents the fitted regression line for predicting delay discounting rate by chronological age using the MM-estimator. In order to prevent overplotting, the points were slightly jittered.

**Figure 2 behavsci-12-00063-f002:**
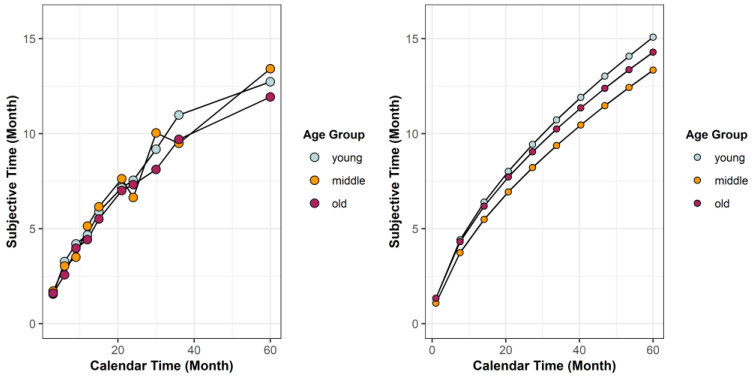
In the left panel, the M-estimate of location of subjective anticipatory time was plotted against calendar time for each group and for each delay. In the right panel, subjective anticipatory time for each delay was estimated using the M-estimate of location of alpha and beta for each age group.

**Figure 3 behavsci-12-00063-f003:**
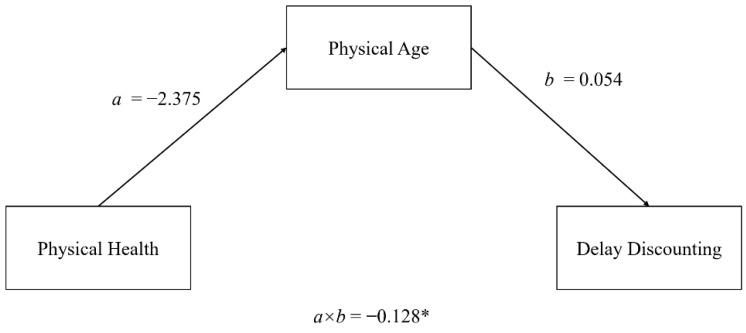
Mediation effects of subjective physical age between physical health and delay discounting. Only the indirect effect was tested. ** p* < 0.05.

**Table 1 behavsci-12-00063-t001:** Participant Characteristics.

Variables	Young Adults	Middle Age	Older Adults
	M (SD)/tau	M (SD)/tau	M (SD)/tau
Sample size	84	54	104
Gender	31M/53F	26M/28F	61M/43F
Chronological age	24.00 (3.79)	39.70 (2.71)	58.80 (3.53)
Physical age	24.20 (4.17)	37.60 (5.82)	53.30 (7.97)
Psychological age	25.40 (5.73)	36.10 (10.3)	47.70 (11.00)
Physical health	3.92 (0.88)	3.59 (1.00)	3.86 (0.81)
Psychological health	3.88 (0.96)	3.67 (1.03)	4.25 (0.80)
Childhood SES	3.73 (1.49)	3.17 (1.40)	3.59 (1.59)
Life history strategy	5.18 (0.80)	4.73 (0.83)	5.40 (0.72)
FTP	0.12	−0.18	−0.07
Delay discounting	−0.23 **	−0.03	0.15
alpha	−0.23 **	0.08	0.02
beta	0.20 *	−0.25 *	0.007

FTP = future time perspective; alpha = overall time contraction; beta = diminishing sensitivity. For the last four rows, robust associations (Kendall’s tau) between chronological age and corresponding variables were calculated by age group. * *p* < 0.05, ** *p* < 0.01.

**Table 2 behavsci-12-00063-t002:** Robust Regression Estimates Predicting Delay Discounting.

Variables	Model 1	Model 2
Intercept	1.18 (1.39)	1.82 (1.38)
Life history strategy	−0.57 * (0.22)	−0.55 * (0.21)
Chronological age	−0.11 (0.06)	−0.14 * (0.06)
Age group	−11.56 ** (4.22)	−12.32 ** (4.16)
Age *Age group	0.26 ** (0.09)	0.30 ** (0.09)
Age bias score		−0.11 * (0.06)
Age bias score *Age group		0.14 * (0.06)
Observations	185	185
*R* ^2^	0.13	0.15

Regression coefficients were shown, with standard errors in parentheses. Age group was dummy coded (0 = young adults, 1 = older adults). ** p* < 0.05, ** *p* < 0.01. In Model 1, we examined the divergent effects of age and life history strategy on delay discounting. Model 2 aimed to explore the underlying mechanism of the age effect. The model examined the impact of physical age on delay discounting by testing the interaction between age bias score (physical age minus chronological age) and age group.

## Data Availability

The data of the current study are available from the corresponding author by request.

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
