# Peer review of "Differential Effects of Fundamental and Longitudinal Life History Trade-Offs on Delay Discounting: An Evolutionary Framework"

_behavsci, 2022, doi:10.3390/bs12030063_

Round 1

Reviewer 1 Report

This paper is very relevant because it deals with a topic of interest such as the effects of Fundamental and Longitudinal Life-HIsotiry trade-Offs on Delay Discounting. The determinants of delay discounting are of great interest to the scientific community. Moreover, this recent study adds interest by addressing this issue by analyzing the effect of childhood SSE on delay discounting. It also attempts to predict the relationship between aging and delay discounting. Different measures were used for this study: age, subjective health, delay discounting, future time perspective, anticipatory time perception, life history strategy and childhood socioeconomic status. The authors found that slower LHS predicted lower DD; the relationship between chronological age and DD was U-16 shaped; and finally, the effects of age and LHS were differential. 

The main strengths have to do with the chosen subject matter and its interest for developmental psychology. On the other hand, the design of the study is interesting since it takes into account the collection of data from different measures. On the other hand, the theoretical framework of the study presents an interesting review that includes a large number of primary and secondary scientific sources. However, the authors should provide a review of more current works. It is recommended that the percentage of references to publications cited in the last five years be increased to 40%. In this regard, the authors only cite 8 papers from the last 5 years out of a total of 76 references. This revision of more current manuscripts should affect the introduction and the discussion section.

On the other hand, the authors should revise the abstract and indicate the number of participants. Likewise, in the abstract they should indicate the population in which they are recruiting. In the section on participants, they should indicate what incentives were used to recruit participants, if any. They should also indicate whether the ethical criteria for research with human subjects were met. 

Overall, the results presented are very interesting. The analyses performed are shown clearly and in detail. 
The authors correctly use statistical tests. In relation to the qualitative study, the authors present interesting data but there is no methodological basis to support the study. Thus, on what theoretical model of qualitative research are they based? is it grounded theory? This point should be addressed in the methodology section. 

Author Response

Point 1: This paper is very relevant because it deals with a topic of interest such as the effects of Fundamental and Longitudinal Life-HIsotiry trade-Offs on Delay Discounting. The determinants of delay discounting are of great interest to the scientific community. Moreover, this recent study adds interest by addressing this issue by analyzing the effect of childhood SSE on delay discounting. It also attempts to predict the relationship between aging and delay discounting. Different measures were used for this study: age, subjective health, delay discounting, future time perspective, anticipatory time perception, life history strategy and childhood socioeconomic status. The authors found that slower LHS predicted lower DD; the relationship between chronological age and DD was U-16 shaped; and finally, the effects of age and LHS were differential.

The main strengths have to do with the chosen subject matter and its interest for developmental psychology. On the other hand, the design of the study is interesting since it takes into account the collection of data from different measures. On the other hand, the theoretical framework of the study presents an interesting review that includes a large number of primary and secondary scientific sources. However, the authors should provide a review of more current works. It is recommended that the percentage of references to publications cited in the last five years be increased to 40%. In this regard, the authors only cite 8 papers from the last 5 years out of a total of 76 references. This revision of more current manuscripts should affect the introduction and the discussion section..

Response 1: Thank you for providing detailed comments and suggestions. We have added 13 papers published since 2018 to enrich our introduction, methods, and discussion section. We believe that this expansion of more recent work provides a more comprehensive background for our paper. Indeed, we admitted that our review encompassed less recent works. This could be attributed to two reasons. First, the critiques of classical statistics we cited were spawned at the end of the 20th century. Second, studies on age differences in delay discounting dated back to 1994 and few conclusions were warranted since then, making less recent studies focus on this research question. The current study aims to summarize previous findings in this field and thus relevant studies published during the past three decades are cited.

Point 2: On the other hand, the authors should revise the abstract and indicate the number of participants. Likewise, in the abstract they should indicate the population in which they are recruiting. In the section on participants, they should indicate what incentives were used to recruit participants, if any. They should also indicate whether the ethical criteria for research with human subjects were met.

Response 2: All of these details were added into the corresponding parts. However, due to the word limit (200 words) of the abstract, we merely indicated that 242 Chinese participants were recruited. More details about the population were added to the methods section.

Point 3: Overall, the results presented are very interesting. The analyses performed are shown clearly and in detail. The authors correctly use statistical tests. In relation to the qualitative study, the authors present interesting data but there is no methodological basis to support the study. Thus, on what theoretical model of qualitative research are they based? is it grounded theory? This point should be addressed in the methodology section.

Response 3: Thank you for your comments and for the interesting question regarding qualitative research. Actually, the construction of theories and hypotheses in the current paper follows the well-established tradition of evolutionary psychology. There are several levels of hierarchy in the theory construction process. Theory of evolution (proposed by Darwin) occupies the highest level in the hierarchy. Then, middle-level theories (e.g., life history theory in our paper) are derived from theory of evolution. Specific hypotheses and predictions are then generated by these middle-level theories in evolutionary biology (there are more details in Evolutionary Psychology: The New Science of the Mind (5ed) by David Buss, Figure 2.1). Researchers then test these falsifiable predictions using psychological data. From this perspective, the theoretical model of qualitative research in the current paper might be the hypothetico-deductive model (e.g., deductive reasoning) rather than grounded theory (e.g., inductive reasoning). However, we are not sure whether the hypothetico-deductive model is a theoretical model of qualitative research. Although we have proposed a theoretical framework, this paper is more like a quantitative study. Following your suggestion, we added some details in the methods section (section 2.3 Analytical Strategy) to explain the theoretical model of qualitative research and how our statistical analyses associate with the theoretical framework.

Reviewer 2 Report

The paper presents a study investigating the effects of life history strategies on delay discounting behaviours using an evolutionary perspective. Two theories underlie their hypotheses: the life history theory and the antagonistic pleiotropy. The results indicated that slower life history strategies predicted lower delay discounting behaviours and a U-shaped relationship between chronological age and delay discounting behaviours.

General judgment comments

The paper brings together two theories in looking at delay discounting in the response to ecological cues and delay discounting behaviour across the lifespan. The introduction was detailed in providing the background to the theories. However, there are some methodological issues that need to be addressed.  

Major issues

1) Methods and analysis for delay discounting

  • It was unclear how delay discounting was scored for each participant. The text only mentioned that participants would make decisions between an immediate reward and a delayed reward.
  • Although the authors pointed out the issue with “classical statistics”, it was unclear why they used Kendall’s tau as they did not report the normality of the data.
  • Figure 1: The authors mentioned using the MM-estimator instead of ordinary least squares regression – however, Figure 1 depicts a line fitted by ordinary least squares.
  • The authors mentioned that they dummy coded age group for the regression of delay discounting on life history strategy, chronological age and age group. However, there was no dummy code for the middle-aged adults.

2) Mediation models

Section 3.4

Although the authors reported “a significant indirect effect of of childhood SES on delay discounting via life history strategy” (Line 346), it did not seem as though childhood SES was included in the models reported in Table 2. Model 1 and 2 reported in Table 2 were not defined clearly in the text either and it was unclear what each model was specifically testing.

  • Table 2: It is not immediately clear what the values in brackets represent (standard error of the regression coefficients?) – it would be helpful to specify what the contents of the table are.

Section 3.5

  • The coefficients of the indirect effects were not reported for the middle aged and young adults.
  • It would be helpful to include a figure of the mediation model for the older adults.

3) Discussion

  • In discussing the results, the authors pointed out that “the magnitude of money” or the reward needs to be considered “fitness relevant”. While they pointed out that “the age-related change in discounting rate may reflect lower marginal utility of rewards for wealthier people”, they did not seem to account for this difference in wealth and marginal utility of rewards (possibly, in terms of the current SES of participants) in the study.

Minor issues

1) It would be good to proofread the paper for language errors.

Author Response

Point 1: It was unclear how delay discounting was scored for each participant. The text only mentioned that participants would make decisions between an immediate reward and a delayed reward.

Response 1: Thank you for providing thoughtful and detailed comments and suggestions. We have added more details about how delay discounting was indexed in the methods section (e.g., indexed by a hyperbolic discount parameter k and how to explain this parameter).

Point 2: Although the authors pointed out the issue with “classical statistics”, it was unclear why they used Kendall’s tau as they did not report the normality of the data.

Response 2: Following your suggestion, we tested the normality using Shapiro-Wilk’s test and reported the results in section 2.3. Analytical Strategy. The results indicated a severe departure from the normal distribution.

Point 3: Figure 1: The authors mentioned using the MM-estimator instead of ordinary least squares regression – however, Figure 1 depicts a line fitted by ordinary least squares.

Response 3: We redrew the scatter plot and added fitted regression lines using the MM-estimator. Note that we randomly jittered the points to prevent from overplotting.

Point 4: The authors mentioned that they dummy coded age group for the regression of delay discounting on life history strategy, chronological age and age group. However, there was no dummy code for the middle-aged adults.

Response 4: Indeed, only young and older adults were coded. We only compared these two groups because the age effect was more evident for these two groups both theoretically and empirically (as shown in section 3.2). According to our theoretical framework, the change in fertility is more pronounced around sexual maturity and at older ages. Thus, the contrast between these two periods (young and older adults) deserves more attention.

Point 5: Although the authors reported “a significant indirect effect of of childhood SES on delay discounting via life history strategy” (Line 346), it did not seem as though childhood SES was included in the models reported in Table 2. Model 1 and 2 reported in Table 2 were not defined clearly in the text either and it was unclear what each model was specifically testing.

Response 5: Table 2 reports two models concerning the differential effects of fundamental and longitudinal life history trade-offs (Model 1) and whether subjective physical age has an incremental effect on delay discounting (Model 2). We added notes below Table 2 and the main focus of these two models in the article to make our results more clear.

Point 6: Table 2: It is not immediately clear what the values in brackets represent (standard error of the regression coefficients?) – it would be helpful to specify what the contents of the table are.

Response 6: The values in brackets represent standard error of estimates. We have added a note below the table to clarify.

Point 7: The coefficients of the indirect effects were not reported for the middle aged and young adults. It would be helpful to include a figure of the mediation model for the older adults.

Response 7: We have reported indirect effects for young adults and middle-aged. A figure of the mediation model was also added.

Point 8: In discussing the results, the authors pointed out that “the magnitude of money” or the reward needs to be considered “fitness relevant”. While they pointed out that “the age-related change in discounting rate may reflect lower marginal utility of rewards for wealthier people”, they did not seem to account for this difference in wealth and marginal utility of rewards (possibly, in terms of the current SES of participants) in the study.

Response 8: We agree that this is a potential limitation of our study and have acknowledged it in Sections 4.3 and 5. An extra reference to this issue was also added. However, this potential confounding effect of current SES is unlikely to significantly affect our conclusions. We argue that the marginal utility of rewards becomes more salient when the rewards are small, which is not the case of our tasks. 

Point 9: It would be good to proofread the paper for language errors.

Response 9: Thank you for the reminder. We have proofread the article and fixed some errors. 

Reviewer 3 Report

Thank you very much for your work. This manuscript has been written by using a good structure. All the main ideas are included on it. The theory is adequate, the method is described properly, the results are in line with the objective and the discussion is correct. Thank you.

What is the main question addressed by the research? Is it relevant and interesting?

- The main question addressed by the study "Differential Effects of Fundamental and Longitudinal Life-History Trade-Offs on Delay Discounting: An Evolutionary Framework" refers to the repercussions or associations that can be found when investigating longitudinal trade-offs in the life history of individuals by studying them with respect to variables such as age or childhood.  - It is interesting because it proposes an approach to the question through the "antagonistic pleiotropy hypothesis".   How original is the topic? What does it add to the subject area compared with other published material? - In relation to the originality of the topic, it should again be noted that trying to respond to the objective of the study by resorting to the hypothesis set out in the previous question is original.  - The analysis of the trade-offs and factors associated with life history and age is extremely useful in a society in which the age of reproduction is lengthening with the following connotations and repercussions that this fact has.   Is the paper well written? Is the text clear and easy to read? Are the conclusions consistent with the evidence and arguments presented? Do they address the main question posed? - The manuscript describes the subject matter including theoretical background, methodology, results, discussion and conclusions. Perhaps, in a second revision of the manuscript, it would be interesting to use the third person singular, instead of the first person plural, to achieve greater objectivity in the writing of the text.  - The conclusions are consistent with the arguments put forward and the study focuses on the main issue raised.   

Author Response

Point 1: Thank you very much for your work. This manuscript has been written by using a good structure. All the main ideas are included on it. The theory is adequate, the method is described properly, the results are in line with the objective and the discussion is correct. Thank you.

What is the main question addressed by the research? Is it relevant and interesting?

- The main question addressed by the study "Differential Effects of Fundamental and Longitudinal Life-History Trade-Offs on Delay Discounting: An Evolutionary Framework" refers to the repercussions or associations that can be found when investigating longitudinal trade-offs in the life history of individuals by studying them with respect to variables such as age or childhood.  - It is interesting because it proposes an approach to the question through the "antagonistic pleiotropy hypothesis".   How original is the topic? What does it add to the subject area compared with other published material? - In relation to the originality of the topic, it should again be noted that trying to respond to the objective of the study by resorting to the hypothesis set out in the previous question is original.  - The analysis of the trade-offs and factors associated with life history and age is extremely useful in a society in which the age of reproduction is lengthening with the following connotations and repercussions that this fact has.   Is the paper well written? Is the text clear and easy to read? Are the conclusions consistent with the evidence and arguments presented? Do they address the main question posed? - The manuscript describes the subject matter including theoretical background, methodology, results, discussion and conclusions. Perhaps, in a second revision of the manuscript, it would be interesting to use the third person singular, instead of the first person plural, to achieve greater objectivity in the writing of the text.  - The conclusions are consistent with the arguments put forward and the study focuses on the main issue raised. 

Response 1: Thank you very much for your comments. We have further checked and improved the article.

Round 2

Reviewer 1 Report

Dear authors, thank you very much for making the suggested modifications. I believe that the manuscript has improved considerably. I endorsed the article.

Reviewer 2 Report

Good study